# The Uncomfortable Truth: Open Thoracotomy versus Minimally Invasive Surgery in Lung Cancer: A Systematic Review and Meta-Analysis

**DOI:** 10.3390/cancers15092630

**Published:** 2023-05-05

**Authors:** Dohun Kim, Wongi Woo, Jae Il Shin, Sungsoo Lee

**Affiliations:** 1Department of Thoracic and Cardiovascular Surgery, College of Medicine, Chungbuk National University Hospital, Chungbuk National University, Cheongju 28644, Republic of Korea; mwille@chungbuk.ac.kr; 2Department of Thoracic and Cardiovascular Surgery, Gangnam Severance Hospital, Yonsei University College of Medicine, Seoul 03722, Republic of Korea; woopendo@gmail.com; 3Department of Pediatrics, Yonsei University College of Medicine, 50 Yonsei-ro, Seodaemun-gu, Seoul 03722, Republic of Korea

**Keywords:** lung cancer, minimally invasive surgery, open thoracotomy, randomized controlled trials

## Abstract

**Simple Summary:**

Surgery is the preferred treatment of choice for early-stage lung cancer, but historically, it required large incisions (open surgery) for access and removal of the tumor. Latest guidelines recommend minimally invasive surgery (MIS) as a preferred option for lung cancer due to better clinical outcomes. However, there is insufficient RCT (randomized clinical trial) evidence to establish the superiority of MIS over open surgery. This study revealed selection bias and problems related to surgical approach, with no significant difference in major postoperative complications between thoracotomy and MIS. Therefore, it is reasonable to allow experienced surgeons to choose the appropriate surgical approach for each patient.

**Abstract:**

For decades, lung surgery in thoracic cancer has evolved in two ways: saving more parenchyma and being minimally invasive. Saving parenchyma is a fundamental principle of surgery. However, minimally invasive surgery (MIS) is a matter of approach, so it has to do with advances in surgical techniques and tools. For example, MIS has become possible with the introduction of VATS (video-assisted thoracic surgery), and the development of tools has extended the indication of MIS. Especially, RATS (robot-assisted thoracic surgery) improved the quality of life for patients and the ergonomics of doctors. However, the dichotomous idea that the MIS is new and right but the open thoracotomy is old and useless may be inappropriate. In fact, MIS is exactly the same as a classic thoracotomy in that it removes the mass/parenchyma containing cancer and mediastinal lymph nodes. Therefore, in this study, we compare randomized-controlled trials about open thoracotomy and MIS to find out which surgical method is more helpful.

## 1. Introduction

The development of thoracic oncologic surgery has progressed from more to less invasive techniques [1]. For example, in the case of lung cancer, the standard extent of removal has shifted from pneumonectomy or bilobectomy to lobectomy or segmentectomy [2,3]. As for the approach or incisions, they were from full thoracotomy, anterolateral, or posterolateral thoracotomy to video-assisted thoracic surgery (VATS) or robot-assisted thoracic surgery (RATS) [4]. Although it may not necessarily be a superior procedure from an oncological standpoint, when a long-lasting scar is replaced by a few small holes and the number of holes decreases further over time, such a procedure is referred to as a progression in surgical techniques [5,6]. However, it is important to remember that such advancements must align with the primary goal of surgery, which is to increase patient survival by curing the disease. If a difficult surgery is performed with less invasive techniques, it may be a great accomplishment for the surgeon, but if it does not meet the appropriate goal of the surgery, it cannot be considered a great achievement for the patient. This is the reason why surgeons tend to approach new treatment methods more conservatively.

At that point, VATS had been validated worldwide for a long time [7,8,9,10,11,12,13,14,15,16,17,18,19]. Many leading groups in the field of thoracic surgery in various countries have applied VATS to more surgeries, proving its safety and effectiveness and expanding its indications since its initial introduction. Recently, RATS has been introduced, which is similar to VATS in that it performs surgery through several holes instead of incisions in the patient’s body [20,21,22,23,24,25,26,27]. The movement of the surgical tools for the RATS is more flexible than VATS, so minute manipulation is easier, and surgeons could do a more ergonomic surgery in the chair, but it is similar to VATS considering outcomes [22,24,25]. When VATS and RATS are combined, it can be defined as minimally invasive surgery (MIS). Initially, during the introduction of MIS, efforts were made to prove that it was not inferior to an open thoracotomy [9,10,13]. Concerns arose that MIS could have limited visibility and handling capabilities compared to an open thoracotomy, which has accumulated various experiences. However, according to the published results, MIS is not only non-inferior, but in some cases, it shows better results [7,11,16]. Therefore, recent guidelines recommend MIS for lung cancer surgery, but it is required to be validated systematically [28,29].

Surgeons also need to approach our clinical guidelines from the perspective of evidence-based medicine (EBM). Though most guidelines value the results of randomized-controlled trials (RCTs), the contemporary EBM approach does not evaluate the certainty or level of evidence by study design alone. To ensure reproducibility of study findings, further evaluation for specific components such as design, risk of bias, or publication bias should be conducted by expert groups. Even several RCT results could point in different directions, so surgeons need to evaluate RCTs related to surgical procedures through systematic reviews and meta-analyses. 

Of course, if MIS is right and open thoracotomy has significant flaws, it should no longer be practiced, and the current attitude should be reinforced. However, to do so, efforts are needed to objectively look at existing research results and look behind the results. It is necessary to confirm under what conditions the advantages and disadvantages of each surgery are justified. Therefore, the purpose of this paper is to review the guidelines for the surgery of thoracic oncology, objectively analyze prospective RCTs (randomized controlled trials) on MIS, and compare the clinical significance of MIS and open thoracotomy in terms of early clinical outcomes.

## 2. Materials and Methods

### 2.1. Review of the Literature

Guidelines were collected and analyzed by the two experienced thoracic surgeons (D.K. and S.L.). Four guidelines were selected as the statements for the MIS and classic thoracotomy (or open) were clear and comparable [28,29,30,31]. Their reputations and citations were also considered for the selection, and then they were summarized as the type of cancer, indication of the surgical treatment, and recommended surgical extent, including lung parenchyma and mediastinal lymph nodes. 

A search of PubMed literature published in English was performed using the following medical terms: ((open) OR (thoracotomy)) AND ((MIS) OR (minimally invasive) OR (VATS) OR (Video)) AND (lung cancer) AND ((Randomized Controlled Trial) OR (Randomized) OR (Randomized) OR (Randomization) OR (Randomization)). Seven RCTs with comparable data on thoracotomy and VATS were finally included in this analysis. We compared preoperative demographics, tumor size, early mortality, and postoperative complications such as bleeding, prolonged air leakage, respiratory failure, and arrhythmia. Two authors (W.W. and J.I.S.) extracted data from article texts, tables, figures, and Appendix A. They independently reviewed and evaluated the quality of each study, and any discrepancies between them were resolved by a thorough discussion with two other authors (D.K. and S.L.). This study was not registered in PROSPERO, and the PRISMA checklist was applied to evaluate this article (Figure 1). 

### 2.2. Statistical Analysis

Tumor size, patient age, operating time, and hospital stays were compared by standardized mean difference. Postoperative complications after thoracotomy or VATS were compared using the RR and 95% confidence interval (CI). I^2^ statistics were used to evaluate heterogeneity, and I^2^ > 50% was considered to represent significant heterogeneity. Due to the low heterogeneity among studies, fixed effect models were used to demonstrate each comparison between VATS and thoracotomy. Statistical significance was defined as a two-sided *p*-value < 0.05. Statistical analyses were performed using R version 4.1.0 (R Foundation for Statistical Computing, Vienna, Austria) and Review Manager (RevMan) software version 5.2.3 (The Nordic Cochrane Centre, Copenhagen, Denmark).

### 2.3. GRADE Approach

Version 2 of the Cochrane risk of bias tool (ROB 2) for randomized studies was used for the six RCTs included in this study. Two surgical experts (S.L. and D.K.) and two specialists (J.I.S. and W.W.) with systematic reviews independently evaluated the risk of biases. Then, we used the GRADE (Grading of Recommendations, Assessment, Development, and Evaluations) approach to evaluate the certainty of evidence based on RCTs.

## 3. Results

### 3.1. Collective Review of the Guidelines

To collect opinions on minimally invasive surgery (MIS) for lung cancers, considering awareness and timing of presentation, the following four guidelines were selected: ACCP 2013, ESMO 2017, NICE 2019, and NCCN 2023 in Table 1. All these guidelines present separate recommendations for small-cell lung cancer (SCLC) and non-small-cell lung cancer (NSCLC). Although there are some differences in the indications for surgery, all the guidelines generally aim for stage I–IIIa lung cancer. There was no disagreement on the indication for surgery according to the staging, as stages I–II were included, but there were differences in the recommendations for stage IIIa. The extent of surgery was not disputed, with lobectomy being the standard operation; however, recent guidelines have recommended the need for anatomic segmentectomy in some early stages with ground-glass opacity [28,29]. According to the guideline, sampling of mediastinal lymph nodes is sufficient rather than mediastinal lymph node dissection [28]. 

Opinions on video-assisted thoracic surgery (VATS), a type of MIS, varied according to the guidelines. ACCP 2013 preferred MIS over thoracotomy for anatomic pulmonary resection in stage I patients and suggested it in experienced centers [31]. However, the 2017 ESMO guidelines stated that either open thoracotomy or VATS could be used as appropriate to the expertise of the surgeon, although they recommended VATS as the appropriate approach for stage I [29]. Both guidelines, however, marked the level of evidence as 2C or V, C, indicating insufficient evidence and a lack of strong recommendations. According to the 2019 NICE guideline, both MIS and thoracotomy could be used for lung cancer surgery, but the 2023 NCCN guideline emphasized that MIS (VATS or RATS) should be strongly considered if the principles of surgery are applied, without specifying the stage [28,30]. In addition, it described excellent early results in terms of pain, hospitalization period, daily recovery, and complications in high-volume centers. In summary, early guidelines in the early 2010s allowed a choice between MIS and thoracotomy, limited to early-stage lung cancer, but recent guidelines recommend MIS, including VATS, for general lung cancer surgery without limitations in staging. However, considering various limitations, it is understandable what the guidelines aim to recommend. Nevertheless, it is questionable whether the evidence is clear.

### 3.2. Randomized Controlled Trials (RCTs) Comparing Open vs. VATS

Seven randomized controlled trials were included in this study, which compared open lobectomy (whether through posterolateral or anterior thoracotomy) versus VATS lobectomy. The inclusion and exclusion criteria used to allocate patients were reviewed. The clinical outcomes of each study were analyzed using meta-analysis methods.

#### 3.2.1. Patients

Review of the patient allocation and surgery

In 1995, Kirby et al. reported the results of a randomized controlled trial (RCT) on 55 patients (n = 15 in VATS and n = 30 in open) [13]. Due to the difficulty of the surgery, three cases had to be converted to thoracotomies, but they were excluded from the final analysis. From the perspective of intention to treat, the exclusion of these cases from the VATS group may have introduced bias in the clinical outcome assessment. On the other hand, some patients in the open group had such difficult surgeries (i.e., with a high possibility of poor clinical outcomes) that bronchial tears occurred during the dissection process and more than 500 mL of blood loss was observed, making it difficult to ensure that both groups had equivalent surgical difficulties. Furthermore, patients with stage II or higher were 36% in open and 20% in VATS.

Surgery: open (muscle sparing posterolateral thoracotomy) vs. VATS (no rib spreading but including 6–8 cm thoracotomy) 

Limitations: unequal assignment of patients with different surgical difficulties and disease stages.

In 2000, Sugi et al. analyzed long-term results (3- and 5-year survival rates) by comparing 100 patients (n = 52 in VATS and n = 48 in open) [9]. Two cases had to be converted to thoracotomy for bleeding control, and they were included in the open group. Additionally, patients with T2 or higher were only present in the open group.

Surgery: open (posterolateral thoracotomy) vs. VATS (8 cm axillary incision with two or three ports)

Limitations: unequal assignment of patients with different surgical difficulties and disease stages.

In 2001, Craig et al. measured the levels of C-reactive protein (CRP), interleukin-6 (IL-6), tumor necrosis factor (TNF), and reactive oxygen species (ROS) production in the immediate postoperative period of 41 patients (n = 22 in the VATS group and n = 19 in the open group) [14]. Although the study referred to another paper for information on randomization, it was not specifically mentioned how randomization was performed, except that it involved peripheral bronchogenic carcinoma [1]. Both groups included patients with benign diseases (n = 4 in VATS and n = 1 in open) and were not equally allocated in terms of higher stage (stage III was none in VATS but n = 1 in open) and cell type (over six cell types were mixed). Interestingly, the VATS group included four patients with benign masses as well as patients with carcinoid, renal metastasis, melanoma metastasis, and high-grade sarcoma, for whom the surgical procedure performed may differ from that in general lung cancer patients, but there was no specific description of whether the surgeries performed on these patients were the same as those in general lung cancer patients.

Surgery: open (posterolateral thoracotomy) vs. VATS (4–5 cm incision with three ports)

Limitations: patients with metastatic and benign tumors, who are expected to have different principles of surgery, are included in the VATS group, and the allocation of disease stage is not equal. 

In 2013, Palade et al. conducted an RCT with 64 patients (n = 32 in VATS and n = 32 in open) [10]. The primary endpoint was the number of dissected lymph nodes. Two cases of conversion were excluded from the VATS group, while two cases of reoperation in the open group were not excluded from the study. Both conversion cases occurred on the left side, and even when including these cases, the proportion of left-sided cases in the VATS group was lower than in the open group (38% vs. 47%). Additionally, one patient with a carcinoid was included in the VATS group but not in the open group.

Surgery: open (anterolateral thoracotomy) vs. VATS (one utility incision; 3–5 cm and two holes)

Limitations: Unequal surgical difficulty of patients, staging, and cancer type.

In 2016, Bendixen et al. conducted an RCT to measure pain score and quality of life (QOL) in 206 patients (n = 103 in VATS and open anterior thoracotomy) over a period of 6 years [8]. Initially, 772 patients were screened, of whom 411 did not meet the inclusion criteria, 69 declared non-participation in the study, and 86 were not asked about their willingness to participate (for unknown reasons) and were thus excluded. One case of conversion to thoracotomy was included in the open group for analysis. Overall, randomization was fair and clear.

Surgery: open (anterolateral thoracotomy) vs. VATS (one utility incision; 4 cm and three holes)

Limitations: There were 86 patients who were not asked about their willingness to participate in the study (which represents 29% of the total patients if they had all participated), and 411 patients were deemed ineligible for the study, but the specific criteria for this determination were not provided, which would have been helpful for reference in other surgical groups.

In 2018, Long et al. published the results of a large RCT that measured safety and short-term efficacy in 425 patients [11]. After randomization, 481 patients were allocated in a 1:1 ratio, with 236 in the VATS group and 245 in the open group. Eight cases of conversion to thoracotomy, including two cases of bleeding and two cases of severe adhesions, were excluded from the study. There were no pneumonectomy patients in the VATS group, while three were in the open group. The pathological results showed that the tumors in the open group were significantly larger, and there was an uneven stage distribution (stage II/III: 25% in VATS and 37% in open). The authors stated that tumor size was not used for randomization, but it is possible that the higher staging in the open group was due to the more thorough lymph node dissection, leading to the discovery of unexpected lymph node metastases, or due to the higher staging of the excluded patients who underwent conversion.

Surgery: open (muscle sparing thoracotomy under the axilla) vs. VATS (not determined but 3–4 holes and utility incision under 5 cm) 

Limitations: Unequal surgical difficulty and disease severity assignment. Differences in the proportion of high-risk pneumonectomy cases with a higher mortality rate.

In 2022, Lim et al. published an RCT with 503 patients (n = 247 in VATS, n = 256 in open) about physical functions at 5 weeks postoperative [7]. They screened 2109 patients and excluded 1606 patients, including 171 for planned wedge, 108 for planned segmentectomy, and 60 for planned pneumonectomy, without explaining why they were allocated to the procedure. The median follow-up was 12.1 months. Most of the ethnicity of the study populations was white (96.8% in VATS and 96.1% in open). They also excluded 313 patients because they had previous malignancies that influenced their life expectancy. However, it would be better to explain how status influences life because patients with previous malignancies were included in the study (about 11% in VATS and 14% in open) and differences between the two groups were not presented. Interestingly, trainee surgeons were included in the study, with more in the open group (11.8% in VATS and 21% in open). This discrepancy could change the results in each group because experienced surgeons, not trainees, could respond well to the unexpected events during the surgery and have better outcomes.

Surgery: open (any thoracotomy) vs. VATS (one to four holes, without rib spreading)

Limitations: not standardized surgical methods. Vague reasons for exclusions of patients and uneven allocation of trainee surgeons (especially in the open group)
Analysis of the patients’ characteristics (Table 2)

Age, sex, and histologic diagnosis were not significantly different between the two groups (Appendix A). However, patients in the VATS group had significantly smaller tumors than the thoracotomy group (standardized mean difference (SMD) −3.4 mm, 95% CI −5.0 mm to −1.8 mm, I^2^ = 6%) (Appendix A). 

#### 3.2.2. Outcomes

Figure 2 and Table 3 describe the early clinical outcome according to two surgical approaches. They did not differ in operating time (SMD 0.29 min, 95% CI −0.06 to 0.64 min, I^2^ = 87%, k = 6) or early mortality (RR 0.71, 95% CI 0.21–2.35, I^2^ = 0%, k = 4). However, hospital stay was significantly shorter in VATS than thoracotomy (SMD −0.18 day, 95% CI −0.29 to −0.07 day, I^2^ = 0%, k = 6).

**Table 3 cancers-15-02630-t003:** Clinical outcome and postoperative complications in each randomized clinical trial.

Study	Operative and Clinical Outcome						Postoperative Complication											
	Early Mortality ⸹		Operating Time (min)		Hospital Stay (Days)		Hemorrhage		PAL ¶		Pneumonia		Arrhythmia		Respiratory Failure ⁂		Chylothorax	
	MIS	Open	MIS	Open	MIS	Open	MIS	Open	MIS	Open	MIS	Open	MIS	Open	MIS	Open	MIS	Open
Kirby et al., (1995) [13]			mean 161 SD 61	mean 175 SD 93	mean 7.1 SD 5.5	mean 8.3 SD 5.7			3/25	8/30								
Sugi et al., (2000) [9]																		
Craig et al., (2001) [14]			mean 141 min (SD 39.5)	mean 121 min (SD 31.4)	mean 8.6 (SD 3.02)	mean 7.9 (SD 3.23)	1 of 22						0/22	1/19	1/22	2/19		
Palade et al., (2013) [10]	1/32	0/32	mean 187 SD 38	mean 158 SD 39	median 9 range 6–25	median 11 range 8–36	0/32	0/32	1/32	1/32	8/32	2/32	1/32	1/32	1/32	4/32	0/32	2/32
Bendixen et al., (2016) [8]	1/102	1/99	median 100 (IQR 80–115)	median 79 min (IQR 60–101)	median 4 days (IQR 2–13)	median 5 days (IQR 2–18)	14/102 ⸙	9/99 ⸙	5/102	6/99			1/102	1/99				
Long et al., (2018) [11]	0/215	0/210	median 150 (IQR 115–195)	median 166 (IQR 130–205)	median 14 (IQR 12–19)	median 15 (IQR 13–19)	2/215	0/210			3/215	5/210	3/215	7/210	1/215	1/210	0/215	2/210
Lim et al., (2022) [7]	2/221 †	5/232 †	median 150 (IQR 120–186)	median 132 (IQR 108–168)	median 4 (IQR 3–7)	median 5 (IQR 3–8)	2/247	2/255	20/135	11/146	37/247	53/255	23/247	22/255	12/247	12/255	0/247	3/255

IQR, interquartile range; MIS, minimally invasive surgery; PAL, prolonged air leakage; SD, standard deviation; † Among patients who underwent lobectomy; ⸹ Mortality during hospitalization for surgery or within 30 days of surgery; ⸙ Requiring reoperation for bleeding; ¶ Air-leakage over 7 days; ⁂ Requiring mechanical ventilation.

**Figure 2 cancers-15-02630-f002:**
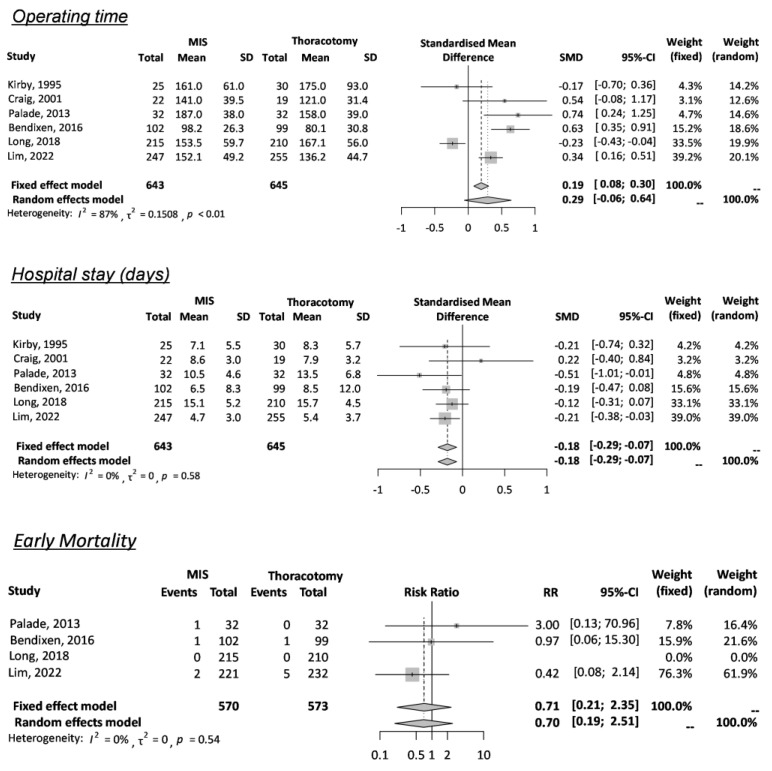
Early clinical outcomes [7,8,10,11,12,13,14].

Six RCTs reported postoperative complications (PoC), and three RCTs specifically classified the adverse degree of PoC; their results were combined and meta-analyzed. When two surgical approaches (VATS and thoracotomy) were compared in each type of PoC, there was no difference in hemorrhage (RR 1.62, 95% CI 0.82–3.22, I^2^ = 0%, k = 5), prolonged air leakage (RR 1.20, 95% CI 0.73–1.99, I^2^ = 39%, k = 4), acute respiratory failure (RR 0.80, 95% CI 0.41–1.54, I^2^ = 0%, k = 4), and arrhythmia (RR 0.89, 95% CI 0.55–1.44, I^2^ = 0%, k = 5) (Figure 3). Though only two RCTs reported a higher level of postoperative pain as a complication, there was no preference between the two approaches (RR 0.83, 95% CI 0.59–1.17, I^2^ = 0%, k = 2) (Appendix A). The GRADE approach evaluated the certainty of evidence, as shown in Table 4. Due to the limited number of studies and variations in measurements, the level of certainty was low (⨁⨁◯◯) for early mortality, hemorrhage, and respiratory failure and moderate (⨁⨁⨁◯) for prolonged air leakage and arrhythmia.

## 4. Discussion

There appears to be no significant benefit to using MIS over open (classic) thoracotomy when considering surgical options for lung cancer, except for a shorter hospital stay. According to meta-analysis, we have not found any statistically significant differences in terms of operating time, early mortality, or complications including hemorrhage, prolonged air leakage, respiratory failure, arrhythmia, or postoperative pain between the two approaches. It is important to note that the majority of research on MIS has focused on stage I patients, and tumors included in MIS are generally smaller in size. Therefore, when interpreting the clinical findings and conclusions from these studies, it is crucial not to overstate their significance.

The conclusions of the above-mentioned RCT are as follows [7,8,9,10,11,13,14]. Kirby et al. reported that VATS was not associated with significant benefits [13]. Similarly, Sugi et al. stated that there were no statistical differences between the MIS and the open for overall survival [9]. In the study by Carig et al., they measured perioperative CRP and IL-6, which were lower in VATS, so they concluded that it would be helpful for decreasing recurrence [14]. However, the values had become similar and were almost equal at postoperative hour 200. Moreover, even without considering allocation issues that could disadvantage the open group, the surgery time and mean length of hospital stay were better in the open group. Importantly, it would be difficult to say whether these parameters could influence tumor recurrence and long-term survival because there is no long-term serologic data or biological background. Palade et al. claimed that MIS was as effective as open for mediastinal lymph node dissection (MLND) and that MIS had a better field of vision [10]. However, as pointed out in the discussion at the conference [10], when MLND is performed using VATS instruments, the possibility of overestimation due to fragmentation should be considered. Bendixen et al. concluded that VATS should be the preferred surgical approach for stage I lung cancer because it showed better value for pain and QOL [8]. However, the continuous variable of pain score was dichotomized into moderate to severe (NRS ≥ 3) or not and analyzed as a categorical variable. Moreover, when pain scores were separated into severe (NRS > 7) and non-severe groups, the proportions of patients did not differ between groups during 52 weeks of follow-up (*p* = 0.17). As the authors noted, they did not obtain complete pain data, and only self-reported QOL was superior in the VATS group. However, most individual dimensions on the EQ5D and EORTC QLQ-C30 questionnaires did not differ significantly between groups at most time points. Moreover, the open group had a shorter surgery time, and all reoperations due to bleeding occurred in the VATS group. Long et al. reported that VATS had an advantage in surgery time and bleeding, but short-term complications did not differ [11]. Despite the allocation issue mentioned earlier, it is important to note that there were no statistically significant differences in chest tube duration, length of hospital stay, or complications between the two groups. Lim et al. concluded that MIS is associated with better recovery of physical function at 5 weeks, but physical function at 6 and 12 months did not differ between the two groups [7]. Although short-term pain scores appeared to be better in MIS, safety issues such as prolonged air leakage and vascular injury were more common in the MIS group, and oncologic outcomes did not differ between the two groups.

According to recent guidelines, MIS is more recommended [28,29], but this may not be appropriate from the following perspectives: First, as previously analyzed, there is insufficient evidence. In addition, in the conclusions of the seven RCTs, three stated that MIS and open are not different [9,10,13], and the remaining four claimed advantages of MIS in limited areas or time points of clinical outcomes, excluding survival [7,8,11,14]. Given the lack of significant benefits for survival, especially in the long term, there is insufficient evidence to prefer a particular approach. Second, it may interfere with surgical decision making. MIS inevitably results in conversion to thoracotomy [32,33,34,35]. Considering that even high-volume centers report significant conversion cases, it is thought to be due to the disease itself and individual physical status rather than a problem with the technique [33,34]. Therefore, when the purpose of surgery is to treat the cancer and improve survival, surgeons can decide on the appropriate timing for conversion based on their experience and judgment. However, in situations where MIS is more recommended by the guidelines, surgeons may obsess over MIS to avoid the non-recommended open approach, which can compromise clinical outcomes due to prolonged surgery time and increased bleeding [32,36]. Third, it can hinder the rapport between surgeons and patients. Patients who experience unavoidable conversion may think that they received surgery that was contrary to the guideline, which can lead medical professionals to feel guilty about not providing the best treatment [37]. This can also have a negative impact on the long-term survival of patients [38,39]. Lastly, recommending MIS can accelerate medical inequality. If two approaches have equal value, individuals will choose cost-effective treatments based on their economic means. However, recommending MIS as the preferred treatment option without considering its limited accessibility in low-income countries and populations [26,40,41,42,43], as well as the limited evidence for its benefits, could result in unreasonable medical inequalities. Moreover, assuming that MIS is less costly than open surgery, as some studies suggest [40,44], vulnerable groups may be compelled to choose expensive and less-valued treatments (i.e., open surgery).

There is no concrete evidence to suggest whether VATS or open mediastinal lymph node dissection is superior to the other. Most agree that it is useful to thoroughly examine mediastinal lymph nodes [45,46]. However, there is disagreement regarding whether to perform sampling or dissection, and there are concerns regarding MIS, such as whether fragmentation occurs more frequently due to the characteristics of the instruments, whether a sufficient number of lymph nodes can be gathered at various stations according to the guidelines, whether it has an impact on survival rates, or whether it affects up- or down-staging after surgery [10,47]. In related studies, there is the above-mentioned RCT by Palade et al. and a nationwide study by Van der Woude et al. Both studies showed no statistically significant difference between procedures. Especially, Van der Woude et al. compared 5154 MIS patients with 2306 open patients and found the following results [47]. First, mediastinal lymph node dissection following international guidance is a minority, and there is no difference in the completeness of mediastinal lymph node dissection between procedures. Second, up- or down-staging after surgery was more common in the open group. Possible reasons for this include selection bias due to retrospective studies and an uneven distribution of stages. However, it should not be overlooked that the open group effectively detected occult lymph node metastases as expected, especially in those with higher T stages. Further studies that are precisely designed are required, as there may be differences in survival rates depending on how mediastinal lymph nodes are managed [45,48].

## 5. Conclusions

The choice between MIS and open thoracotomy should be considered value-neutral until decisive evidence emerges from the perspectives of oncologic concern and survival. The decision to choose either method should be left to the judgment of the surgeons who have selected lung cancer surgery as their main focus, and the basis of the decision should prioritize maximum patient safety and survival. Of course, the advancement of lung cancer surgery should be minimally invasive, and in this regard, we are following a proper process. However, such efforts should not undervalue proven surgical methods without sufficient evidence. It is important not to forget that the primary value of lung cancer surgery is to treat the cancer and improve survival rates.

## Figures and Tables

**Figure 1 cancers-15-02630-f001:**
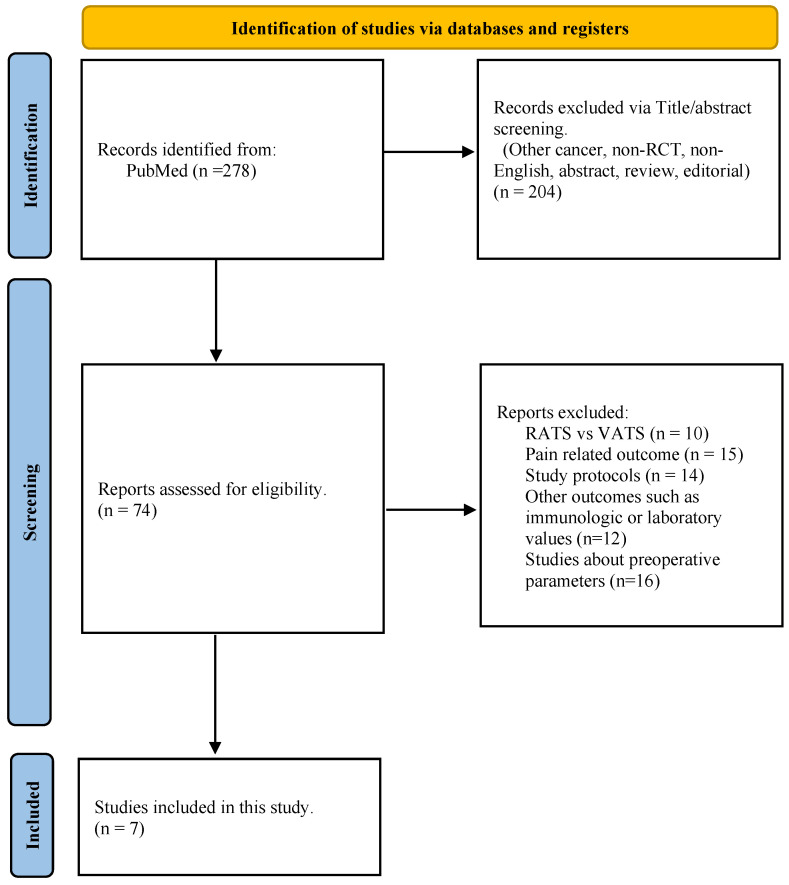
PRISMA flow diagram.

**Figure 3 cancers-15-02630-f003:**
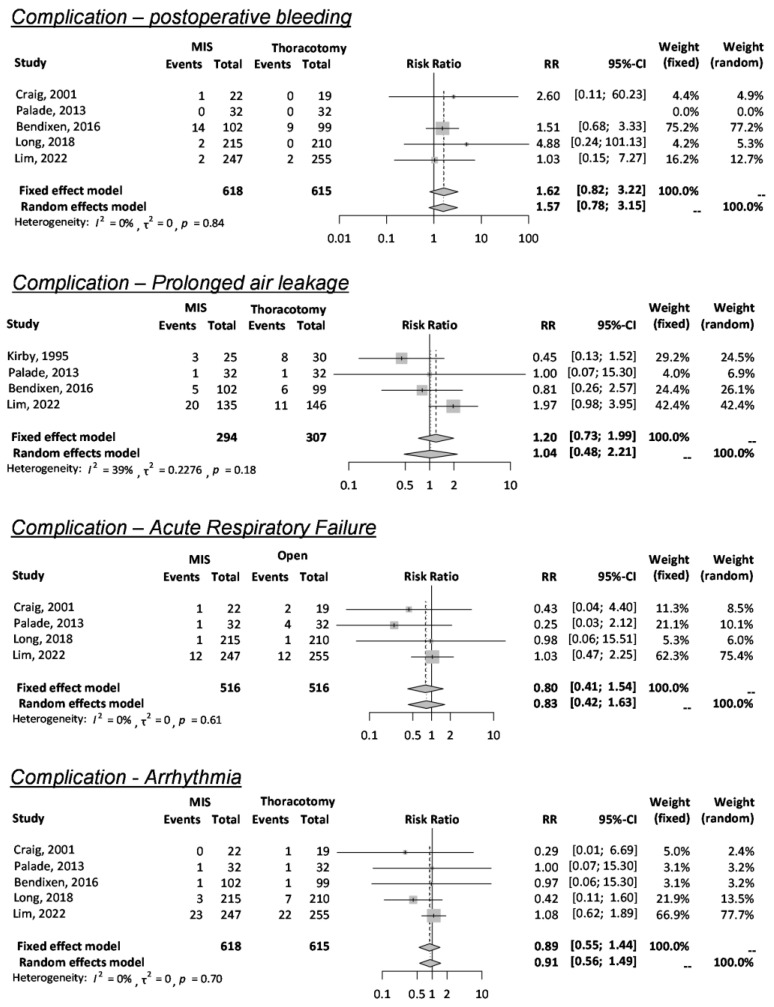
Postoperative complications [7,8,10,11,12,13,14].

**Table 1 cancers-15-02630-t001:** Guidelines for thoracic oncology was summarized focusing on the indication of MIS for lung cancer surgery.

Name	Indication of Surgery	Surgical Extent	Mediastinal Lymph Node	Opinions for MIS
**Treatment of stage I and II NSCLC; Diagnosis and management of lung cancer, 3rd edition: ACCP guidelines** [2]	-Stage I and II	-For clinical stage I and II, a lobectomy is recommended -For stage I predominantly GGO less than 2 cm sublobar resection with negative margin is suggested	-SLND (stage II) or SLNS (stage I) is required	-For stage I, a MIS (VATS) is preferred over a thoracotomy for anatomic pulmonary resection and is suggested in experienced centers (Grade 2C; weak recommendation with low level of evidence).
**ESMO Clinical Practice guidelines (2017)** [3]	-Stage I and II -Single N2 (with neoadjuvant or adjuvant) -Superior sulcus or resectable T3/T4 (with neoadjuvant)	-Lobectomy is still considered the standard operation of tumors over 2 cm -Segmentectomy for pure GGO, AIS, or MIA	-SLND is recommended in Stage II and IIIA	-Either open thoracotomy or VATS access can be carried out as appropriate to the expertise of the surgeon. -Standard open thoracotomy or VATS is probably less important from oncologic perspective. -VATS should be the approach of choice in stage I tumors (V, C; studies without control group, case reports, expert opinions with insufficient evidence for efficacy.
**NICE Lung cancer: diagnosis and management (March 2019)** [4]	-For whom are well enough and for whom treatment with curative intent is suitable	-Offer lobectomy (either open or thoracoscopic)	-SLNS or SLND	-Either open or thoracoscopic
**NCCN Guidelines: NSCLC (version 2, 2023)** [5]	-Stage I and II -Stage IIIA (N2): for resectable cases -T3 (invasion) and T4 local extension tumors	-Anatomic pulmonary resection is preferred. -Segmentectomy (preferred) or wedge is appropriate in selected	-SLNS	-VATS or MIS (including RATS) should be strongly considered for patients with no anatomic or surgical contraindications. -In high-volume centers with significant VATS experience, VATS lobectomy in selected patients results in improved early outcomes (decreased pain, reduced length of hospital stays, more rapid return to function, fewer complications) without compromise of cancer outcomes. -RATS seems to be more expensive than conventional VATS.

MIS, Minimally Invasive Surgery; NSCLC, Non-small cell lung cancer; ACCP, American College of Chest Physicians; GGO, Ground-Glass Opacity; SLND, Systemic Lymph Node Dissection; SLNS, Systemic Lymph Node Sampling; VATS, Video-Assisted Thoracic Surgery; ESMO, European Society for Medical Oncology; AIS, Adenocarcinoma In situ; MIA, Minimally Invasive Adenocarcinoma; NICE, National Institute for Health and Care Excellence; NCCN, National Comprehensive Cancer Network; RATS, Robotic-Assisted Thoracic Surgery.

**Table 2 cancers-15-02630-t002:** Patients’ characteristics of included randomized clinical trials.

Author (Year)	Country	Period	Inclusion Criteria	Number of Patients			Excluded for Analysis	Male		Age		Adenocarcinoma		Tumor Size	
				Total Enrolled	MIS Analyzed	Open Analyzed		MIS	Open	MIS	Open	MIS	Open	MIS	Open
Kirby et al., (1995) [13]	USA	1991–1993	Clinical stage Ⅰ	61	25	30	Non-malignant lesion,					13/25	13/30		
Sugi et al., (2000) [9]	Japan	1993–1994	Clinical stage IA	100	48	52		28/48	29/52	mean 65.9 SEM 1.4	mean 64.9 SEM 1.4	non-squamous 33/48	non-squamous 41/52	mean 20.2 mm SEM 1.8	mean 22.6 mm SEM 1.2
Craig et al., (2001) [14]	UK		Peripheral opacity lesions	44	22	19	Patients’ refusal	8/22	14/19	median 64.5 (range 46–78)	median 62 (range 47–74)	178/215	161/210		
Palade et al., (2013) [10]	Germany	2008–2011	Clinical stage Ⅰ	66	32	32		21/32	23/32	mean 67.7 SD 8.6	mean 64.7 SD 7.3	20/32	21/32	mean 23.5 mm SD 14.4	mean 24.3 mm SD 13.8
Bendixen et al., (2016) [8]	Denmark	2008–2014	Clinical stage I	206	102	99	Non-malignant lesion, other malignancy	50/102	47/99	median 66 IQR 62–72	median 65 IQR 60–72	61/102	61/99		
Long et al., (2018) [11]	China	2008–2014	Clinical Stage Ⅰ–Ⅱ	481	215	210	SCLC Non-malignant lesion, age > 75	105/215	107/210	mean 57.11 SD 9.069	mean 58.1 SD 9.22	178/215	161/210	median 25 mm IQR 17–32	median 30 mm IQR 20–40
Lim et al., (2022) [7]	UK	2015–2019	Clinical stage I–III	503	247	255		119/247	130/255	Mean 69 SD 8.7	Mean 69 SD 9.0	80/247	91/255		

IQR, interquartile range; MIS, minimally invasive surgery; SCLC, small cell lung cancer; SD, standard deviation; SEM, standard error of mean.

**Table 4 cancers-15-02630-t004:** GRADE table describing the quality of evidence and importance of recommendations.

Certainty Assessment							№ of Patients		Effect		Certainty	Importance
№ of Studies	Study Design	Risk of Bias	Inconsistency	Indirectness	Imprecision	Other Considerations	VATS	Thoracotomy	Relative (95% CI)	Absolute (95% CI)		
**Early mortality**												
4	randomised trials	not serious	not serious	not serious	very serious ^a^	none	4/570 (0.7%)	6/573 (1.0%)	**RR 0.71** (0.21 to 2.35)	**3 fewer per 1000** (From 8 fewer to 14 more)	⨁⨁◯◯ Low	CRITICAL
**Hemorrhage**												
5	randomised trials	serious	not serious	not serious	serious ^a^	Different indication for intervention	19/618 (3.1%)	11/615 (1.8%)	**RR 1.62** (0.82 to 3.22)	**11 more per 1000** (from 3 fewer to 40 more)	⨁⨁◯◯ Low	CRITICAL
**Prolonged air-leakage**												
4	randomised trials	not serious	not serious	not serious	serious ^a^	Different treatments for air-leakage	29/294 (9.9%)	26/307 (8.5%)	**RR 1.20** (0.73 to 1.99)	**17 more per 1000** (from 23 fewer to 84 more)	⨁⨁⨁◯ Moderate	IMPORTANT
**Respiratory Failure**												
4	randomised trials	serious	not serious	not serious	serious ^a^	none	15/516 (2.9%)	19/516 (3.7%)	**RR 0.80** (0.41 to 1.54)	**7 fewer per 1000** (from 22 fewer to 20 more)	⨁⨁◯◯ Low	IMPORTANT
**Arrhythmia**												
5	randomised trials	not serious	not serious	not serious	serious ^a^	none	28/618 (4.5%)	32/615 (5.2%)	**RR 0.89** (0.55 to 1.44)	**6 fewer per 1000** (from 23 fewer to 23 more)	⨁⨁⨁◯ Moderate	IMPORTANT

CI: confidence interval; RR: risk ratio; a. Due to the limited number of cases.

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
