# Peer review of "The Uncomfortable Truth: Open Thoracotomy versus Minimally Invasive Surgery in Lung Cancer: A Systematic Review and Meta-Analysis"

_cancers, 2023, doi:10.3390/cancers15092630_

Round 1

Reviewer 1 Report

Thank you for submitting this article. I was pleased to receive it as a reviewer.

I have some queries and suggestions that you should address first.

First, the paper should be written according to the Preferred Reporting Items for Systematic Reviews and Meta-Analyses (PRISMA) statement (http://www.prisma-statement.org/).

Secondly, the manuscript should be registred in PROSPERO.

Authors should declare if they perform a certified librarian search. Authors should add other databases in their search (e.g. CINAHL, EMBASE, Cochrane).

The PICO questions should be declared before the systematic review.

I would read the string used.

The heterogeneity of the studies should be commented in the discussion.

Risk of bias table should be added.

The statistical analysis should be rewritten according to the recently published guidelines (Hickey GL, Dunning J, Seifert B, Sodeck G, Carr MJ, Beyersdorf F on behalf of the EJCTS and ICVTS Editorial Committees Editor’s Choice: Statistical and data reporting guidelines for the European Journal of Cardio-Thoracic Surgery and the Interactive CardioVascular and Thoracic Surgery. Eur J Cardiothorac Surg 2015;48:180-93).

The table should be rewritten according to the suggestions of the paper of Hickley et al.

The discussion should be improved with a better discussion about the limitations of meta-analyses.

About minor points, there are typo errors in the text. Authors should thoroughly check the paper.

The references should be written according to the author’s instructions. 

Good luck with your article, and thanks again for submitting it.

Author Response

First, the paper should be written according to the Preferred Reporting Items for Systematic Reviews and Meta-Analyses (PRISMA) statement (http://www.prisma-statement.org/). Secondly, the manuscript should be registred in PROSPERO. Authors should declare if they perform a certified librarian search. Authors should add other databases in their search (e.g. CINAHL, EMBASE, Cochrane). The PICO questions should be declared before the systematic review. I would read the string used.  The heterogeneity of the studies should be commented in the discussion. Risk of bias table should be added. The statistical analysis should be rewritten according to the recently published guidelines (Hickey GL, Dunning J, Seifert B, Sodeck G, Carr MJ, Beyersdorf F on behalf of the EJCTS and ICVTS Editorial Committees Editor’s Choice: Statistical and data reporting guidelines for the European Journal of Cardio-Thoracic Surgery and the Interactive CardioVascular and Thoracic Surgery. Eur J Cardiothorac Surg 2015;48:180-93). The table should be rewritten according to the suggestions of the paper of Hickley et al. The discussion should be improved with a better discussion about the limitations of meta-analyses.

-->Thank you for your thoughtful comments. We are truly honored to have your suggestions. As you stated, there are several issues that we need to discuss regarding the design of this study. This study is not systematic review article, and it is much closer to the literature review. As you pointed out, there are several structural issues if we evaluate this study as a systematic review article. First, we did not undergo systematic search. We only searched articles through PubMed; therefore, there could be studies missing. When it comes to this topic comparing open and minimally invasive surgery, there are numerous studies; but most of them are retrospectively designed, which have substantial biases in the interpretation of the results. To minimize selection and other biases, we aimed to include RCTs only and meta-analyzed several key outcomes. As previous systematic review article related to this issue included retrospective studies, we believe that our study has unique or balanced results than others. Second, this study was not registered in PROSPERO and did not follow the PRISMA guideline strictly as it was initially designed as a systematic review. Regarding the heterogeneity of the studies, we would add further comments and we think the heterogeneity was not relatively high as I2 in most outcomes did not exceed 50%. We expect our explanation is apprehensible for your evaluation of this study design and our analysis. If you need further explanation and modification, please let us know. We look forward to hearing from you.

About minor points, there are typo errors in the text. Authors should thoroughly check the paper. 

--> Thank you for your kind comment. We checked and revised them.

The references should be written according to the author’s instructions. 

--> Thank you for the comment. The references were presented as the journal's guidelines. 

Reviewer 2 Report

Kim et al. compared randomized-controlled trials regarding open thoracotomy and minimally invasive surgery (either video or root-assisted) to find out which surgical method is more helpful.

The work is quite comprehensive and the results well discussed.

Despite the up-to date e complete data presented I believe a major shortcoming is posed by the lack of results and discussion regarding lymphadenectomy which is only marginally mentioned from a work of Palade in 2013. In light of recent advances about the prognostic significance of complete lymph node dissection (van der Woude L, Wouters MWJM, Hartemink KJ, Heineman DJ, Verhagen AFTM. Completeness of lymph node dissection in patients undergoing minimally invasive- or open surgery for non-small cell lung cancer: A nationwide study. Eur J Surg Oncol. 2021;47(7):1784-1790. doi:10.1016/j.ejso.2020.11.008 Osarogiagbon RU. The Pathologic Nodal Staging Quality Gap: Challenge as Opportunity in Disguise. J Thorac Oncol. 2022;17(11):1247-1249. doi:10.1016/j.jtho.2022.08.004, Survival After Mediastinal Node Dissection, Systematic Sampling, or Neither for Early Stage NSCLC Ray, Meredith A. et al. Journal of Thoracic Oncology, Volume 15, Issue 10, 1670 - 1681 ) I believe the review should be improved with a paragraph discussing feasibility and results of lymph node dissection in open versus minimally invasive thoracic surgery.

Author Response

Despite the up-to date e complete data presented I believe a major shortcoming is posed by the lack of results and discussion regarding lymphadenectomy which is only marginally mentioned from a work of Palade in 2013. In light of recent advances about the prognostic significance of complete lymph node dissection (van der Woude L, Wouters MWJM, Hartemink KJ, Heineman DJ, Verhagen AFTM. Completeness of lymph node dissection in patients undergoing minimally invasive- or open surgery for non-small cell lung cancer: A nationwide study. Eur J Surg Oncol. 2021;47(7):1784-1790. doi:10.1016/j.ejso.2020.11.008 Osarogiagbon RU. The Pathologic Nodal Staging Quality Gap: Challenge as Opportunity in Disguise. J Thorac Oncol. 2022;17(11):1247-1249. doi:10.1016/j.jtho.2022.08.004, Survival After Mediastinal Node Dissection, Systematic Sampling, or Neither for Early Stage NSCLC Ray, Meredith A. et al. Journal of Thoracic Oncology, Volume 15, Issue 10, 1670 - 1681 ) I believe the review should be improved with a paragraph discussing feasibility and results of lymph node dissection in open versus minimally invasive thoracic surgery.

--> Thank you for your keen and clear comments. We added a paragraph about MLND via VATS or open using the references that you recommended. I think that the paper has been better with your precious comment. 

--> There is no concrete evidence to suggest whether VATS or open mediastinal lymph node dissection is superior to the other. Most agree that it is useful to thoroughly examine mediastinal lymph nodes. [45, 46]However, there is disagreement regarding whether to perform sampling or dissection, and there are concerns regarding MIS, such as whether fragmentation occurs more frequently due to the characteristics of the instruments, whether a sufficient number of lymph nodes can be gathered at various stations according to the guidelines, whether it has an impact on survival rates, or it affects up- or down-staging after surgery.[10, 47] In related studies, there is an above mentioned RCT by Palade et al. and a nationwide study by Van der Woude et al. Both studies showed no statistically significant difference between procedures. Especially, Van der Woude et al. compared 5,154 MIS patients with 2,306 open patients and found the following results.[47] First, mediastinal lymph node dissection following international guidance is a minority, and there is no difference in the completeness of mediastinal lymph node dissection between procedures. Second, up- or down-staging after surgery was more common in the open group. Possible reasons for this include selection bias due to retrospective studies and uneven distribution of stage. However, it should not be overlooked that the open group effectively detected occult lymph node metastases as expected, especially in those with higher T stages. Further studies that are precisely designed are required as there may be differences in survival rates depending on how mediastinal lymph nodes are managed. [45, 48]

Round 2

Reviewer 1 Report

The paper should be written according to the Preferred Reporting Items for Systematic Reviews and Meta-Analyses (PRISMA) statement (http://www.prisma-statement.org/). A PRIMA checklist should be added.

Author Response

Dear Reviewer

Thank you for your sincere review.

We will add PRISMA check list and revised paper.

Please find them.

Thank you again for the review.

Authors

Reviewer 2 Report

I believe the manuscript has been improved and is now suitable for publication.

Author Response

Dear Reviewer

Thank you for your kind and sincere answer.

We'll prepare next steps.

Authors.

Round 3

Reviewer 1 Report

The paper should be written according to the Preferred Reporting Items for Systematic Reviews and Meta-Analyses (PRISMA) statement (http://www.prisma-statement.org/). A PRISMA flow diagram should be added as Figure 1.

Author Response

Thank you for your sincere review for the paper.

The PRISMA flow diagram was added in the Figure 1 and it stated in the main text.

Please find the revised file